# Learning Topology-Agnostic EEG Representations with Geometry-Aware Modeling

**Ke Yi**[*]
South China University of Technology
cs_kerry@mail.scut.edu.cn

**Yansen Wang**[†]
Microsoft Research Asia
yansenwang@microsoft.com

**Kan Ren**[†]
Microsoft Research Asia
kanren@microsoft.com

**Dongsheng Li**
Microsoft Research Asia
dongsli@microsoft.com

## Abstract

Large-scale pre-training has shown great potential to enhance models on downstream tasks in vision and language. Developing similar techniques for scalp electroencephalogram (EEG) is suitable since unlabelled data is plentiful. Meanwhile, various sampling channel selections and inherent structural and spatial information bring challenges and avenues to improve existing pre-training strategies further. In order to break boundaries between different EEG resources and facilitate cross-dataset EEG pre-training, we propose to map all kinds of channel selections to a unified topology. We further introduce **MMM**, a pre-training framework with **M**ulti-dimensional position encoding, **M**ulti-level channel hierarchy, **M**ulti-stage pre-training strategy built on the unified topology to obtain topology-agnostic representations. Experiments demonstrate that our approach yields impressive improvements over previous state-of-the-art techniques on emotional recognition benchmark datasets. [1]

## 1 Introduction

Scalp electroencephalography (EEG), a physiological signal that shows the human brain's status and can be collected in a non-invasive and low-cost manner, is widely used in disease diagnosis, medical monitoring, rehabilitation, and some recent brain-computer interface systems [1, 2, 3]. Although collecting EEG signals is relatively easy, interpreting the signals and providing meaningful labels often require huge expert efforts. Therefore, a few recent works have emerged to learn representations of EEG segments with self-supervised learning methods and apply the pre-trained model to several downstream tasks, including emotion recognition, sleep staging, epileptic seizure detection, and so on [4, 5, 6, 7, 8]. While there are many publicly available EEG datasets for pre-training, transferring across them is very challenging because these EEG recordings can vary significantly in terms of montage (the number and the places of electrodes placed on the scalp) and sample rate. These factors will completely change the format and the EEG contents' meaning. For example, a typical setting of the emotion recognition task involves 62 channels based on the 10-10 electrode system to track the fine-grained changes with very high spatial resolution, and a more practical setting to monitor seizure events may only have 19 channels based on the 10-20 system. Nevertheless, previous self-supervised learning methods avoid handling this problem and thus can only work within a single dataset, which hinders the model from being trained on more available resources.

---

[*]The work was conducted during Ke Yi's internship at Microsoft Research Asia.

[†]Correspondence to Yansen Wang and Kan Ren.

[1]Project link: https://seqml.github.io/MMM/.

37th Conference on Neural Information Processing Systems (NeurIPS 2023).

Another challenge of EEG pre-training is to encode the structure and model the relationships among EEG signals from different locations. These signals are collected by the electrodes placed on the scalp, which can be viewed as a 2-D manifold with spatial information. This spatial information is often crucial for the expert to provide precise labels. For example, lesions can be located by comparing the amplitudes and phases of abnormal discharges between the left and right brain, and changes in emotional states may be reflected by asymmetrical EEG activity [9, 10, 11].

In summary, to get a capable and powerful EEG pre-trained model, we need 1) well encoding spatial information into the representation and 2) pre-training models with EEG corpus having various sensor configurations. In this paper, we propose **MMM**, a self-supervised learning framework based on the masked auto-encoder [12], which maps EEG data with different sensor configurations into unified representations. We introduce region-wise tokens to extract local information from the EEG channels, where region-wise tokens and origin channels form a **M**ulti-level hierarchy learning. The well-learned region-wise tokens form the unified representation, which can be applied in downstream tasks with rich spatial information. We demonstrate that introducing **M**ulti-dimensional position encoding to encode geometric information of sensors benefits pre-training, and **M**ulti-stage mask strategy with random mask and region-wise mask can enhance the hidden representation's robustness. To further evaluate the effectiveness of our proposed framework, we validate our approach on emotion recognition tasks, which is a typical EEG downstream task and requires the model to capture the spatial information well.

We summarize our contributions as follows:

- We propose to use a unified topology to model the scalp and map different montages onto the same topology.

- We propose a novel pre-training framework to learn unified geometry-aware EEG representations on top of the unified topology that can be generalized to different EEG channel configurations.

- Our pre-trained model gains state-of-the-art performance on the emotion recognition benchmarks. More experiments demonstrated its strong ability to transfer between different datasets even with different montages.

## 2   Related Works

**Self-supervised learning for EEG**   Electroencephalograms (EEGs) are commonly used to diagnose neurological, psychiatric, and sleep disorders and in applications involving brain-machine interfaces. In the EEG domain, self-supervised learning has emerged as a promising approach. In [8], the author designs two pretext tasks, temporal context prediction, and contrastive predictive coding, to perform representation learning of EEG signals and applies them to two downstream tasks: sleep staging and pathology detection. [6] introduce a set of data augmentations for EEG and extend the SimCLR framework to extract channel-wise features on time-series EEG data. The above methods focus on temporal information modeling while ignoring spatial information. [13, 14] attempt to solve the cross-montage problem by selecting a fixed channel set, while it might cause spatial information loss when faced with different configurations. [7] use GNN and RNN to model the spatiotemporal dependencies while using the generative pre-training method to improve model performance further. However, their GNN-based model assumes a fixed channel set in their experiments and might naturally be worse at transferring than the attention-based model.

To the best of our knowledge, there is no method trying to use cross-dataset EEG as their pre-training corpus.

**EEG-based emotion recognition**   Before the deep learning methods were extensively adopted in this area, spectral EEG features were commonly investigated, such as power spectral density (PSD), differential entropy (DE), and differential asymmetry (DASM), from which DE feature is proved to be the most precise and stable one in EEG-based emotion recognition tasks [11]. Several EEG-based emotion recognition methods have been proposed based on the DE feature. [15, 16, 17] apply recurrent neural networks to traverse signals and capture the spatial dependency between the EEG channel. However, RNN-based modeling takes the EEG channel as a fixed sequence, which loses sensors' geometric information and makes transferring hard. [18, 19] apply graph neural network to model spatial relationship among EEG channel and GCN-based model achieve state-of-the-art in

EEG-based emotion recognition. However, these GCN-based models are only claimed to be used in supervised training. [4] has shown that Masked EEG modeling is an effective self-supervised pre-training paradigm for EEG data. However, this work takes EEG data as a sequence while ignoring the sensors' physical location information. Again, the above methods are all not trying to pre-training/training on cross-dataset EEG.

# 3 Methodology

## 3.1 Task Formulation

Learning a good EEG representation involves modeling from both temporal and spatial aspects. While temporal modeling is better explored in the previous works, our main focus is on spatial structures. We use DE feature [20] extracted from the raw EEG to aggregate the temporal information. This simplification can bring several benefits: 1) DE feature is a well-studied feature that is demonstrated to be effective for temporal modeling of EEG, especially in EEG-based emotion recognition tasks [11, 21]. 2) The process of DE feature extraction doesn't involve learnable parameters and can exclude the eliminating impact of sample rate variations. This allows the feature to be seamlessly transferred across datasets with different time resolutions. 3) The extracted features are widely used across a wide range of baseline models [22, 23, 24], thereby facilitating equitable comparisons among various frameworks. We describe how DE features are extracted in the Supplementary file. It is worth noting that our proposed method is also applicable to raw EEG signals and other kinds of features as long as the temporal information is handled in a consistent manner across datasets.

In this way, the EEG pre-training task can be formulated as learning a unified hidden representation $h = \text{Encoder}(x) \in \mathbb{R}^H$ from a dataset $\mathcal{X}$ containing the extracted DE features $x \in \mathcal{X} \subset \mathbb{R}^{C_x \times F}$, which can be afterward used in downstream tasks. $H$ is the dimension of the hidden representations. $C_x$ is the number of channels corresponding to a set of sensors $\mathcal{E}_x = \{E_1, E_2, ..., E_{C_x}\}$ used to collect the signal. $F$ is the dimension of the DE feature, which specifically equals 5 in our case, representing the DE feature extracted on five frequency bands, i.e., $\delta$ band (1-4Hz), $\theta$ band (4-8Hz), $\alpha$ band (8-14Hz), $\beta$ band (14-31Hz), $\gamma$ band (31-50Hz).

Unlike vision pre-training, EEG data within a single task is limited, while cross-task EEG data varies significantly in the sensor configuration. Our target is to 1) pre-train a general feature extractor for EEG signals on an irregular dataset $\mathcal{X}$, which breaks the constraint that data have to share the same sensor configuration, and 2) encode the geometric information of the sensors properly which reflect the locations of the EEG signals collected.

## 3.2 Unified Topology

The core idea of addressing the problems is to map different sensors' configurations onto a unified topology and extract a generalized representation that relies on the geometric information of sensors. Recent research in neuroscience reveals that the biological nervous system forms several functional groups in different brain regions [25], and these regions are interconnected with physical, functional, and effective connectivity [26]. These functional regions provide us with a plausible basis for modeling EEG signals which provide a macroscopic view of brain activities.

Drawing inspiration from the functional organization of the brain, we abstract the scalp as a unified topology that can be segmented into several regions $\mathcal{R} = \{R_1, R_2, ..., R_S\}$. Each region can be represented by a vector $z \in \mathbb{R}^D$. Consequently, the whole scalp can be represented by the collection of vectors from all regions as $Z \in \mathbb{R}^{S \times D}$, where $S$ denotes the number of regions and $D$ denotes the dimension of a regional representation. In this paper, we follow the previous findings [19] and divide the scalp into $S = 17$ regions, as shown in Figure 1 (b). We leave the analysis of the influence of different region settings in the supplementary material.

Hence, the challenge of processing data with various montages can be viewed as mapping data $x \in \mathcal{X}$ with the varying sensor configuration $\mathcal{E}_x$ into the representations of the unified topology $Z = \text{Encoder}(x)$. These representations in the same shape can thus be used to transfer among different datasets without any burdens. Moreover, because we design a plausible mapping between channels and regions, the geometric information can be naturally incorporated.

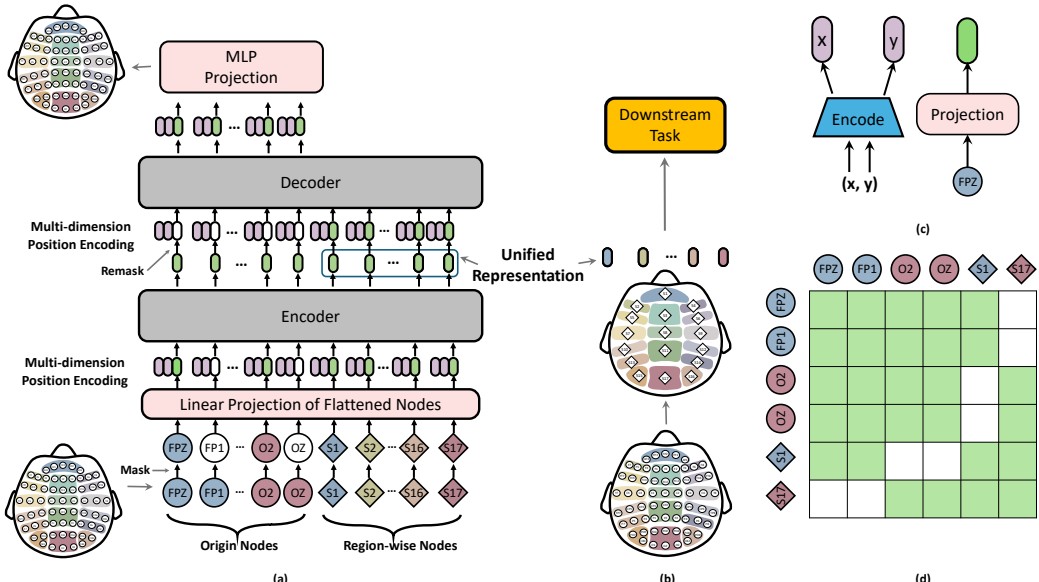

Figure 1: (a) Overview of our methods. Circles and diamonds represent original nodes and region-wise nodes. Nodes of the same color belong to the same region of the unified topology. White patterns get masked/re-masked before being fed into the encoder/decoder. (b) Mapping the original EEG data into a unified topology based on channel locations. (c) Illustration of the multi-dimensional positional encoding. (d) Attention masks. Green means the two tokens interact with attention, and white means no direct interaction.

## 3.3 MMM: an EEG Pre-training Framework on Top of Unified Topology

We introduce MMM, an EEG pre-training framework based on the unified topology that follows the Masked Auto-Encoder (MAE) [12] training schema, i.e., during pre-training, the model first encodes partially-masked irregular input tokens (DE features) to a unified representation with an encoder and then aims to reconstruct the masked tokens from the unified representation with a decoder.

Our designs of MMM to better support EEG pre-training are 3-fold: 1) Multi-dimensional positional encoding, which injects the geometric information into tokens; 2) Multi-level channel hierarchy, where extra tokens representing aggregated regions are added to the original set of channels. Tokens and channels interact hierarchically; 3) Multi-stage pre-training, where two different masking strategies, global random masking and regional masking, are used alternatively to learn the hierarchical representations effectively.

### 3.3.1 Multi-dimensional Positional Encoding

Traditional positional encoding only considers injecting sequence order information. Nevertheless, the injecting order of channels can be quite random and doesn't have actual meanings, and taking channels as a simple sequence will lose the spatial information for the sensors.

As aforementioned, sensors can be conceptualized as placed on a 2-D manifold, so the actual position of each sensor $E$ can be represented as a 2-D coordinate $(x, y)$. We introduce a multi-dimensional positional encoding on this manifold to allow the model to render its learned representations spatially position-aware. The multi-dimensional positional encoding is defined as:

$$x_p = R(x), \quad y_p = R(y),$$
$$\texttt{Encode}(x_p, 2i) = \sin \frac{x_p}{\Omega^{\frac{2i}{d}}}, \quad \texttt{Encode}(x_p, 2i+1) = \cos \frac{x_p}{\Omega^{\frac{2i}{d}}},$$
$$\texttt{Encode}(y_p, 2i) = \sin \frac{y_p}{\Omega^{\frac{2i}{d}}}, \quad \texttt{Encode}(y_p, 2i+1) = \cos \frac{y_p}{\Omega^{\frac{2i}{d}}},$$
$$\texttt{PE}_{x,y,i} = \text{CONCAT}[\texttt{Encode}(x_p, i), \texttt{Encode}(y_p, i)].$$

Here, $(x, y)$ is a 2-D coordinate from the scalp, $d$ is the number of sensors, and $d$ pairs of $(x, y)$ corresponding to the grid of the 10-10 system form a space. Function $R$ projects $x, y$ to their rank

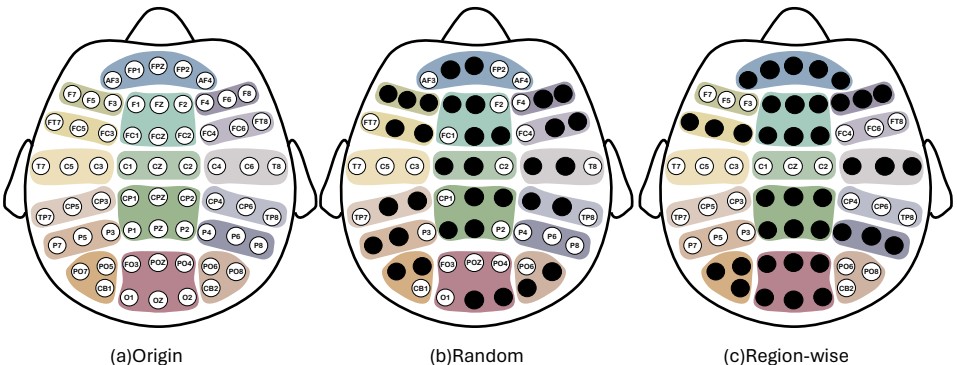

Figure 2: Examples of random mask and region-wise mask. For the region-wise mask strategy, channels in the same group will be all visible or all masked.

value in its original space. The $i$ is the index of feature dimension in the encoding, and $\Omega$ is a large constant (normally set to 10000). The final encoding is generated by concatenating the encoding of the $x$ and $y$ coordinates.

The multi-dimensional positional encoding is used in two places. Firstly, it is added after the DE features from different channels are processed through a linear layer, thereby injecting geometric information into the data. Secondly, the encoding is re-applied after obtaining the representations. Position encoding is performed on the re-masked representations to inform the decoder how to distinguish re-masked tokens.

### 3.3.2 Multi-level Channel Hierarchy

Overall, we employ a transformer-based encoder-decoder as the backbone model of MMM under the masked auto-encoder schema. The encoder-decoder processes representations of tokens with multi-head attention for several rounds. Besides the origin nodes from existing channels, a set of region-wise tokens are accompanied to model the regions of the unified topology.

**Encoder**  Our encoder comprises a stack of $N$ Transformer blocks. Under the pretext of reconstruction, we set a fraction of all tokens to zero and then attempt to reconstruct them with the remaining tokens. We introduce region-wise tokens $s \in \mathbb{R}^{17 \times D}$ and append them to the end of the sequence. This process allows extracting local information from the original channels through the attention mechanism, as shown in Figure 1 (a). In order to tackle potential redundancy that could arise during the interaction of region-wise tokens with nodes from other regions, we assign a fixed attention mask to the encoders/decoders. This ensures precise messages pass between the region-wise nodes and their corresponding origin nodes, as well as amongst region-wise nodes and origin nodes. An example of an attention mask can be seen in Figure 1 (d). The encoder maps a sequence of tokens $x \in \mathbb{R}^{(C+17) \times F}$ to $Z \in \mathbb{R}^{(C+17) \times D}$, where the last 17 tokens form the hidden representation.

**Decoder**  Our decoder is symmetrical to encoder with $N$ Transformer blocks. Tokens from the origin EEG channels are re-masked, and the decoder can only access information contained in the region-wise nodes. The bottleneck design forces the encoder to model EEG well with the unified topology.

### 3.3.3 Multi-stage Pre-training

MMM reconstructs the inputs and predicts the DE value of EEG channels, which consists of the previous $C$ decoder's output tokens. We use the mean square error (MSE) loss to measure the difference between the predicted DE value and the ground truth.

As a new hierarchy is introduced and only the information from region-wise nodes is used to reconstruct the origin nodes, the unified representation must be capable of both intra-region and inter-region reconstruction. On the one hand, every vector in the representation of the unified topology contains aggregated information about a specific region and serves to reconstruct fine-grained features within the region. On the other hand, the interactions between different regions should also be learned in case information from one region is completely missing. Therefore, to further boost the training of

the representations, we introduce the following two masking strategies accordingly and use them in turns during pre-training, as seen in Figure 2, and we call it multi-stage pre-training.

**Global Random Masking**   Channels are masked randomly and independently, and the number of masked channels is fixed. This provides the chance for the region-wise nodes to see partial information within the region and use it to reconstruct the masked intra-region channels.

**Regional Masking**   Channels within the same region are either fully observable or completely masked, and the number of masked regions is fixed. In this way, we force the masked regions to learn representations from region-wise nodes of other regions since they are no longer directly linked to any observable origin nodes with attention.

## 3.4   Apply MMM to Downstream Tasks

To apply MMM to downstream tasks, such as the emotion recognition task, we extract the unified representation of the DE feature and feed it into a 2-layer MLP. As a classification task, the cross-entropy loss is calculated to fine-tune the model. It's noteworthy that the unified representation, which is a hidden vector, can serve in several types of downstream applications.

# 4   Experiments

In this section, we first introduce the datasets we used and the overall settings of our experiments. Then, we compared our model with a series of competitive baselines under different settings to demonstrate the strong modeling and transferring ability between different datasets of our proposed MMM framework. We also include ablation studies to investigate the contributions of the designs separately and showcase the effectiveness of the unified representation for EEG modeling.

## 4.1   Datasets and Experiment Setup

**SEED**   SEED [11] is a discrete EEG emotion dataset elicited by videos. The SEED dataset contains EEG signals of fifteen subjects (eight females and seven males) recorded by 62 EEG sensors while watching fifteen Chinese film clips for three types of emotions. For each subject, the EEG signal recording process is repeated in three different periods corresponding to three sessions. Each session includes the EEG signals while watching the fifteen film clips corresponding to fifteen trials.

**SEED-IV**   SEED-IV dataset [27] is another version of the SEED dataset. It comprises EEG data of fifteen  subjects (eight females and seven males, not overlapping with SEED) recorded in 62 channels. Each subject has three sessions, and each session contains 24 trials.

**SEED-Union**   We combine the SEED and SEED-IV datasets to form the SEED-Union dataset, which is only used for pre-training. Its train set consists of the training samples in SEED and SEED-IV and so as the test set. There is no label in SEED-Union since it's only for the reconstruction pre-training task.

**Lite series**   We evenly select 32 channels from the 62 channels in SEED and SEED-IV datasets to form the SEED-Lite and SEED-IV-Lite datasets. The selected 32 channels are evenly distributed on the scalp. Then, we combine the SEED-Lite and SEED-IV-Lite datasets to form the SEED-Union-Lite dataset, which is also only used for pre-training.

**TUEG**   TUEG [28] dataset consists of clinical recordings using a mostly conventional recording configuration (monopolar electrodes in a 10-20 configuration, sample rate varying from 250Hz to 1024Hz) of 14,987 people for 27,063 hours in total. The subjects were 51% female, and ages ranged from under 1 year old to over 90.

**Experiment setup**   Previous baselines on SEED series datasets all use public DE features, while SEED and SEED-IV provide DE features with different STFT window sizes. It's more reasonable to pre-train the model on the DE features with the same STFT window size. Unless specifically mentioned, the experiments of our model are conducted on the 1-second DE features on both SEED and SEED-IV. Refer to the Appendix B.2 for baseline descriptions and training details.

## 4.2 Subject-dependent Classification

We compare our models in different settings on SEED and SEED-IV with established baselines, and the results are shown in Table 1.

Table 1: Comparison of our method with baseline methods on **SEED** and **SEED-IV** dataset on the subject-dependent classification task. Window size is the length of features to construct the input. For example, 4s× 5 means every DE feature aggregates 4-second raw EEG signals, and five consecutive DE features are used as the model input. The dagger symbol (†) represents the exclusion of the worst-performing session out of three for each subject when calculating the final average accuracy.

| Method | Pre-trained | Window Size | | Accuracy (%) / Std (%) | |
|---|---|---|---|---|---|
| | | SEED | SEED-IV | SEED | SEED-IV |
| STRNN [17] | ✗ | 1s×9 | - | 89.50 / 7.63 † | - |
| DGCNN [18] | ✗ | 1s×1 | 4s×1 | 90.40 / 8.49 † | 69.88 / 16.29 † |
| BiDANN [23] | ✗ | 1s×1 | 4s×1 | 92.38 / 7.04 † | 70.29 / 12.63 † |
| BiHDM [29] | ✗ | 1s×1 | 4s×1 | 93.12 / 6.06 † | 74.35 / 14.09 † |
| R2G-STNN [15] | ✗ | 1s×9 | - | 93.34 / 5.96 † | - |
| RGNN [24] | ✗ | 1s×1 | 4s×1 | 94.24 / 5.95 † | 79.37 / 10.54 † |
| MD-AGCN [22] | ✗ | 1s×5 | 4s×5 | 94.81 / 4.52 | 87.63 / 5.77 |
| MV-SSTMA [4] | ✓ | 1s×10 | 4s×10 | 95.32 / 3.05 | 92.82 / 5.03 |
| V-IAG [30] | ✗ | 1s×1 | - | 95.64 / 5.08 † | - |
| HDGCN [19] | ✗ | 1s×1 | - | 96.40 / 4.54 † | - |
| MAE [12] | ✗ | 1s×1 | 4s×1 | 87.89 / 8.06 | 78.22 / 14.21 |
| MAE [12] | ✓ | 1s×1 | 4s×1 | 91.85 / 7.72 | 81.09 / 10.96 |
| MMM (ours) | ✗ | 1s×1 | 1s×1 | 94.59 / 6.49 | 86.04 / 10.04 |
| MMM (ours) | ✓ | 1s×1 | 1s×1 | **95.76** / 5.53 | 87.92 / 9.24 |
| MMM (ours) | ✓ | 1s×1 | 1s×1 | **97.80** / 3.28 † | 91.78 / 7.05 † |
| MMM (ours) | ✓ | - | 4s×1 | - | **93.62** / 7.74 |

Although our method uses only one-tenth of the information, our method outperforms the previous state-of-the-art self-supervised learning methods, achieving a performance enhancement of 0.44% on the SEED dataset and 0.77% on the SEED-IV dataset. When compared with the supervised methods, our model exhibits a significant improvement, yielding a 1.52% increase in performance on the SEED dataset and a substantial 14.25% enhancement on the SEED-IV dataset, not to mention their setting to exclude the worst-performing session.

After pre-training, MAE and MMM get an accuracy increase of 3.96%/2.87% and 1.17%/1.88% on SEED/SEED-IV, respectively, confirming the significance of EEG pre-training. Moreover, our proposed MMM framework, though following the MAE training schema, outperforms MAE by 3.91%/12.51% on SEED/SEED-IV. This improvement suggests the effectiveness of our specific designs for EEG pre-training.

## 4.3 Transfer between Different Datasets

In this section, we examine the models' transferability across datasets with identical sensor configurations. We pre-train models on the original dataset, including SEED, SEED-IV, and their combinations. Subsequently, the fine-tuning performance of these pre-trained models is evaluated on the target datasets, **SEED** and **SEED-IV**.

As Table 2 demonstrates, all pre-trained variants outperform the corresponding baseline model trained from scratch. Interestingly, the model pre-trained on the SEED-Union by MAE only gets the second-best fine-tuning performance on both datasets. This indicates that even if the channel configuration is identical, the underlying difference in the data distribution among datasets may hinder the model without a unified representation from taking advantage of more data resources. However, while models pre-trained by MMM and MAE share the same architecture, the trend is reversed for models pre-trained by MMM. The model pre-trained with the SEED-Union dataset exhibits the

Table 2: Transfer study: Models get pre-trained on the origin dataset, (-) denotes models get trained from scratch. Then we fine-tune and test the models on the target dataset. The best results for each method and target dataset have been marked in bold.

| Pre-training Dataset | Accuracy (%) | | | |
| | SEED | | SEED-IV | |
| | MMM | MAE | MMM | MAE |
|---|---|---|---|---|
| - | 94.59 | 87.89 | 86.04 | 78.22 |
| SEED | 95.15 | 90.96 | 87.25 | **81.09** |
| SEED-IV | 95.35 | **91.85** | 86.85 | 79.15 |
| SEED-Union | **95.76** | 90.59 | **87.92** | 80.50 |

best downstream performance on both datasets. This indicates that our method, built on the unified topology, can better use a larger number of data to extract more generalizable representations. For more training details, refer to the supplementary.

## 4.4 Transfer between Different Montages

We further investigate the transferability of our method across different sensor configurations in this section. We pre-train models using the Lite series dataset and then conduct a partial fine-tuning on the full SEED dataset, i.e., the parameters of the encoder are frozen, and only the last MLP is tuned. As reflected in Table 3, the model pre-trained by MMM on both the SEED-Lite and SEED-Union-Lite datasets outperforms the model trained from scratch. However, the model pre-trained by MMM on SEED-IV-Lite shows a performance drop. We suggest that the SEED-IV, with only 32 channels, might present a substantial data shift, making it challenging for the pre-trained model to fine-tune/partially fine-tune. However, all models pre-trained using the Mask Autoencoder (MAE) fail to match the partially fine-tuning performance of the model trained from scratch. On the other hand, the model pre-trained using the Multi-modal Multi-task (MMM) approach on the SEED-Union-Lite dataset showcases superior performance. This suggests again that the incorporation of the unified topology aids in the extraction of more generalized representations.

Table 3: Partially fine-tuning results on SEED dataset.

| Pre-trained Dataset | MAE | | MMM | |
| | Acc. (%) | Change | Acc. (%) | Change |
|---|---|---|---|---|
| SEED (Fine-tune) | 91.86 | - | 95.29 | - |
| - | 79.82 | -12.04 | 92.54 | -2.75 |
| SEED | 76.07 | -15.79 | 93.23 | -2.06 |
| SEED-Lite | 74.37 | -17.49 | 92.70 | -2.59 |
| SEED-IV-Lite | 75.80 | -16.06 | 91.92 | -3.37 |
| SEED-Union-Lite | 75.05 | -16.81 | 92.81 | -2.48 |

## 4.5 Transferred from large-scale EEG corpus

Table 4: Large-scale pre-training: Models get pre-trained on TUEG dataset, (-) denotes models get trained from scratch. Then, we fine-tune and test the models on the SEED dataset.

| Pre-training dataset | epoch | Fine-tuning Accuracy (%) |
|---|---|---|
| - | 0 | 94.59 |
| TUEG | 1 | 94.85 |
| TUEG | 2 | 95.11 |
| SEED | 200 | 95.15 |
| TUEG | 3 | 95.29 |

In light of the preceding investigations discovering the effectiveness of our methodology across varying datasets and montages, we proceeded to deploy our approach on TUEG dataset — a large-

scale EEG corpus. Notably, although TUEG dataset collects data on a distinct channel configuration involving 21 channels in the 10-20 system, its utilization in pre-training remains beneficial. As shown in Table 4, there is a discernible increment in the downstream classification accuracy as the number of pre-training epochs increases. Remarkably, when leveraging TUEG for pre-training, our method outperforms self-pre-training within just three epochs. These findings may illuminate the path toward constructing foundation models in the EEG domain.

## 4.6    Impact of Region Division

Table 5: Results from variants of different region division selection

| Number of Regions | Accuracy |
|---|---|
| 1 | 93.95 |
| 4 | 93.41 |
| 7 | 93.68 |
| 17 (chaos) | 93.96 |
| 17 (full attention) | 94.08 |
| 62 | 94.53 |
| 17 (default) | **94.61** |

In this section, we thoroughly analyze the impact of different unified topologies in **MMM** on the SEED dataset. We compared our default 17-region setting with several variants. : 1) 1 single region for all channels; 2) 4 regions shown in Figure 5 (b); 3) 7 regions shown in Figure 5 (d); 4) 62 regions, each channel taking up an individual region; 5) 17 in chaos, where channels are divided to 17 groups without considering spatial relationships. Results are shown in Table 5. From the results, we can conclude that a proper choice of region setting contributes to a better result. Refer to the Appendix C.1 for comprehensive analysis.

## 4.7    Ablation Study

In this section, we scrutinize the design of MMM for both the SEED and SEED-Lite datasets. Table 6 provides an ablation study on the contributions of different components within the MMM model.

Table 6: Ablation study on different components of MMM on the SEED dataset and SEED-Lite dataset.

| Multi-dimensional Positional Encoding | Multi-level Channel Hierarchy | Multi-stage Pre-training | SEED | SEED-Lite |
|---|---|---|---|---|
| | | | 90.96 | 90.74 |
| ✓ | | | 91.46 | 91.31 |
| ✓ | ✓ | | 95.28 | 94.29 |
| ✓ | ✓ | ✓ | 95.15 | 94.62 |

The table reveals that introducing region-wise tokens leads to the most significant enhancement in performance. This underscores the significance of capturing local and regional patterns within the EEG data for effective representation learning.

Moreover, the EEG-specific position encoding (PE) also bolsters the overall performance of the model, as indicated by a 0.4% rise in accuracy. This underlines the value of incorporating spatial information into the model, which aids in understanding the interrelationships among EEG channels and their respective locations on the scalp. Interestingly, pre-training with a multi-stage masking strategy slightly decreases performance on the full-channel SEED dataset but boosts performance on the SEED-Lite dataset. This suggests that the multi-stage mask strategy aids in enhancing the model's generalizability across various datasets. However, it may introduce a slight trade-off, potentially causing minor performance decreases on certain datasets.

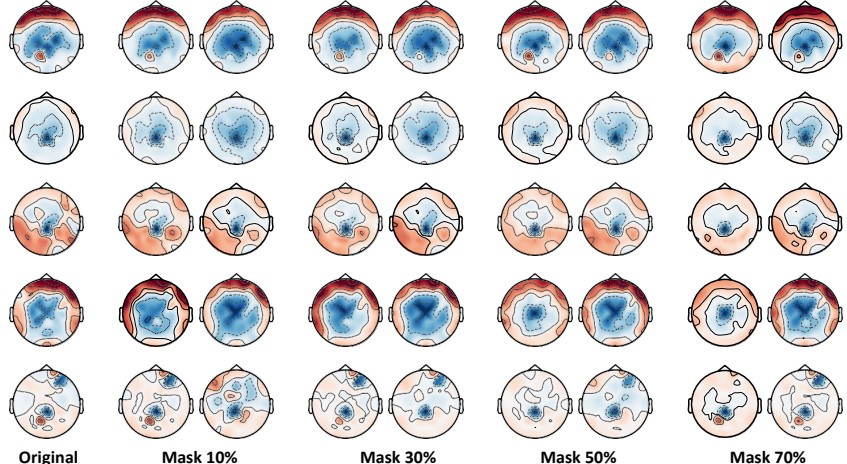

Figure 3: The topographic maps of the reconstructed test data with different mask rates in the $\gamma$ band. The row denotes different test samples from the SEED dataset. The first column is the origin data, while the rest columns stand for the topology reconstructed by MAE pre-trained model (Left) and the topology reconstructed by MMM pre-trained model (Right) with different mask rates.

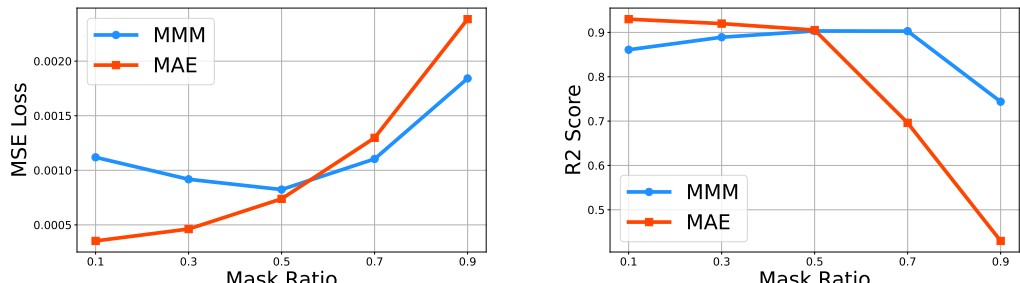

Figure 4: Reconstruction losses on the test samples of SEED

## 4.8    EEG Reconstruction with Unified Representation

Figure 3 presents the visualization of the reconstruction quality for both the mask autoencoder and our method. By analyzing the reconstruction losses shown in Figure 4, we find that MMM demonstrates slightly inferior reconstruction results at a low mask ratio compared to MAE, which can be attributed to the amount of information they feed into reconstructors. Specifically, MAE's reconstructor can access nearly all origin tokens at a low mask ratio, while MMM's reconstructor relies solely on high-level representation, forgoing all origin tokens. However, as the mask ratio increases, MAE's performance starts to collapse while MMM maintains superior reconstruction quality. This phenomenon further substantiates that our approach can effectively model EEG data by the unified topology, yielding robust representations with rich information.

## 5    Conclusion

In this study, we have proposed MMM, an innovative framework for EEG pre-training, which effectively navigates the complexities associated with various sensor configurations in EEG data and well encodes the spatial information of EEG. Our model employs a self-supervised learning approach based on the masked auto-encoder, allowing it to benefit from a vast cross-dataset corpus and enhance the performance of downstream tasks. Our framework demonstrates state-of-the-art performance in emotion recognition tasks, and further evaluations substantiate the efficacy of our methodology. Looking forward, we envision extending our approach to more downstream tasks and modeling temporal and spatial information of EEG data efficiently.

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

## A  DE feature extraction

The key idea of extracting DE features from EEG is calculating the accumulated power within several pre-defined frequency bands. We use $x \in \mathbb{R}^{C \times T}$ to denote the raw EEG data, where $C$ is the number of EEG channels and $T$ is the number of samples (i.e., N = time duration × sampling frequency). The procedure for processing DE features is defined as:

$$w(t) = 0.5 - 0.5 \cos \left( \frac{2\pi t}{N-1} \right), \quad t = 0, 1, ..., N-1, \tag{1}$$

$$s_w(t) = s(t)w(t), \tag{2}$$

$$X(\omega) = FFT\{s_w(t)\}, \tag{3}$$

$$P(k) = |X(k)|^2, \quad k = 0, 1, ..., \frac{N}{2}, \tag{4}$$

$$DE(i) = \frac{1}{n_i} \sum_{k=start_i}^{end_i} P(k), \quad i = 1, 2, ..., F. \tag{5}$$

$n_i$ is the number of samples in the $i$-th frequency band while $start_i$ and $end_i$ marks its start and end point. $F$ is the number of frequency bands. Following medical practices, previous work extract DE feature on five frequency bands, i.e., $\delta$ band (1-3Hz), $\theta$ band (4-7Hz), $\alpha$ band (8-13Hz), $\beta$ band (14-30Hz), $\gamma$ band (31-50Hz). With the help of DE features, one-second-long, $C$-channel EEG data with hundreds of samples can be represented as a $C \times 5$ vector, which can help us extract and aggregate temporal information and avoid inconsistency of sampling frequency in different datasets.

## B  Training Details

Here, we describe the settings used for pre-training and fine-tuning/partially fine-tuning our models on SEED series datasets.

### B.1  Pre-training

We use 6 Transformer layers as encoder and symmetrical layers as decoder. The input EEG data should have 62 channels. Since the EEG data size varies greatly, we first add the channel with random noises. Then, we normalize the EEG data according to the mean and standard deviation calculated on the training set. The hidden dimension of each channel is set to 16. For multi-dimensional positional encoding, we encode x and y with 8 dimensions, respectively, and the combinations are added to the hidden vectors. For *global random masking*, we pick the masking ratio to 0.5. For *regional masking*, we pick the ratio of masking region to 0.35 for the competitive difficulty. For each 100 epochs, the previous 50 epochs take global random masking, and the rest take region-wise masking. For region division, we set 17 regions as mentioned before, i.e., 17 region tokens are appended to the input sequence and used for reconstruction.

We use 1 NVIDIA V100 GPU to train models for 600 epochs with a batch size of 512. We use AdamW to optimize the network. Learning rate and weight decay are grid-searched to the optimal.

### B.2  Fine-tuning

We load the weights of the pre-trained encoder and add a randomly initialized classifier upon the region tokens. In the SEED series, each subject has three sessions, and each session contains multiple trials. For each session, the ahead trails are viewed as training samples, and the rest are viewed as testing samples. The training/test split is 9/6 for SEED and 16/8 for SEED-IV, i.e., for the SEED dataset, each subject contains 27 trials as training samples and 18 trials as testing samples. The final results come from the average of all subjects. We grid-search the learning rate and weight decay, following the same strategy in [4]. We use 1 NVIDIA V100 GPU to train the model for 100 epochs with a batch size of 32.

### B.3 Partially fine-tuning / linear probing

Similar to fine-tuning, we load the pre-trained weight of the pre-trained encoder and add the classifier. However, we freeze the encoder during the backward propagation to conduct partially fine-tuning / linear probing experiments.

### B.4 Baselines

- **STRNN** [17]: The spatial-temporal recurrent neural network is based on a unified spatial-temporal dependency model that learns the information from both spatial and temporal domains of EEG signals.

- **DGCNN** [18]: Dynamical graph convolutional neural network learns the representations of EEG signals by graph convolution in a dynamic way for EEG-based emotion recognition.

- **BiDANN** [23]: Bi-hemispheres domain adversarial neural network focuses on discriminative features of EEG signals from both the right and left sides of the hemispheres of the brain for EEG-based emotion recognition.

- **BiHDM** [16]: Bi-Hemispheres discrepancy model investigates the asymmetric differences of the right and left hemispheres of the brain.

- **R2G-STNN** [15]: A region-to-global spatial-temporal neural network model learns the global and regional EEG representations in both spatial and temporal aspects of EEG signals.

- **RGNN** [24]: Regularized graph neural network explores the topology of EEG channels with graph convolution.

- **MD-AGCN** [22]: A multi-domain adaptive graph convolutional network taking full advantage of features on different domains.

- **MAE** [12]: Masked autoencoders as scalable self-supervised learners by reconstructing the missing patches in images for computer vision.

- **MV-SSTMA** [4]: A spatial-temporal masked autoencoder by reconstructing the missing channels in EEG signals.

## C  Analysis of Different Experiment Settings

### C.1  Impact of Region Division

Table 7: Results from variants of different region division selection

| Number of Regions | Accuracy |
|:---:|:---:|
| 1 | 93.95 |
| 4 | 93.41 |
| 7 | 93.68 |
| 17 (chaos) | 93.96 |
| 17 (full attention) | 94.08 |
| 62 | 94.53 |
| 17 (default) | **94.61** |

Our approach to handling diverse datasets with different montages maps different channel configurations to a unified topology. Our paper uses a 17-region setting based on some neurological findings. However, We would like to investigate how a different selection of regions may influence the model behavior built on it.

In this section, we thoroughly analyze the impact of different unified topologies in **MMM** on the SEED dataset. We compared our default 17-region setting with several variants: 1) 1 single region for all channels; 2) 4 regions shown in Figure 5 (b); 3) 7 regions shown in Figure 5 (d); 4) 62 regions,

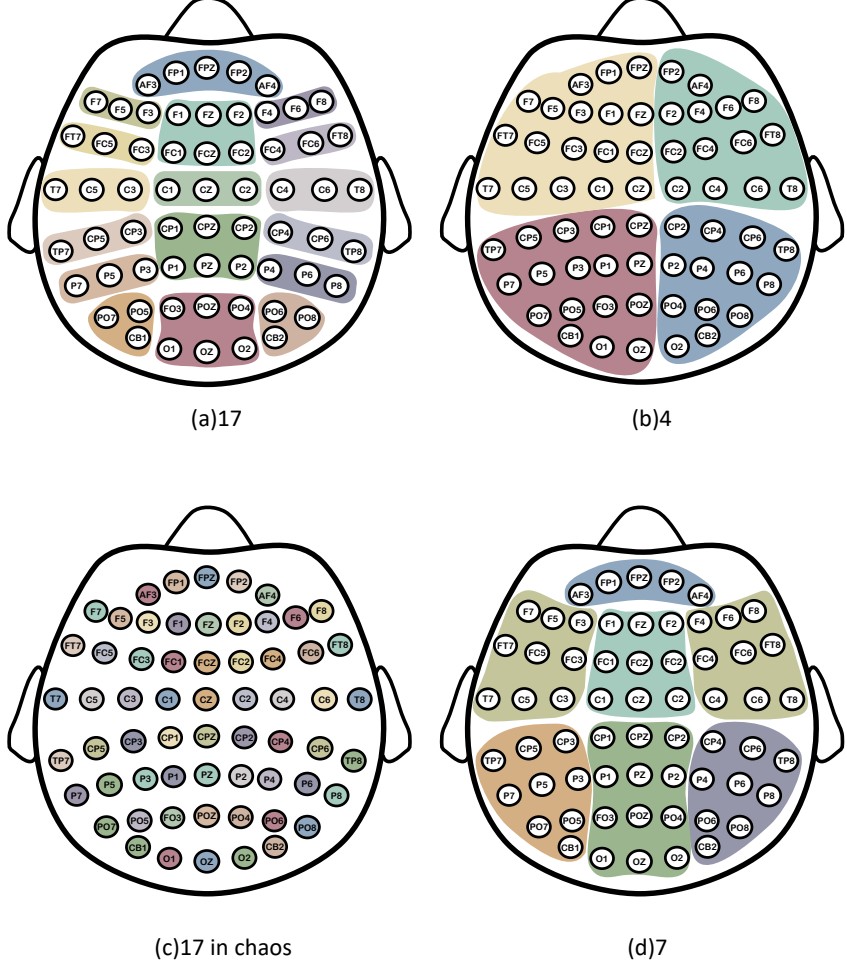

(a)17        (b)4

(c)17 in chaos        (d)7

Figure 5: Different region division

each channel taking up an individual region; 5) 17 in chaos, where channels are divided to 17 groups without considering spatial relationship. Results are shown in Table 7[3].

From the results, we can conclude that a proper choice of region setting contributes to a better result. The selection of the number of regions is a trade-off between modeling fine-grained spatial information and aggregating local information of a larger range. Therefore, in our experiments, a division with fewer regions can only gather coarse-grained spatial information, and more regions may fail to aggregate local information properly, which both perform worse than our default 17-region setting.

Moreover, by comparing our default 17-region setting with the chaos 17-region setting, we can see that the design of aggregating spatial information is also essential to the performance. This suggests that a good unified topology does not mean inserting an appropriate amount of extra tokens but also taking locality and native functionality into consideration when dividing regions.

## C.2 Impact of Masking Ratio

In this section, we investigate the impact of the masking ratio for **MMM** on the SEED dataset.

The global random mask rate is set to different ratios, and for regional masks, the mask ratio is set to have 15% less than the random mask ratio for a proximal difficulty. Results are shown in Table 8

---

[3]Considering the pre-training and downstream evaluation are time-consuming, the space of grid search is smaller for experiments in the Appendix, which may result in a slightly different reported performance.

Table 8: Results from variants of different masking ratio

| Random Masking Ratio | Accuracy |
|:---:|:---:|
| 0.3 | 94.54 |
| 0.5 | **94.61** |
| 0.7 | 94.40 |
| 0.9 | 94.57 |

As we expect, a lower masking ratio results in a weak pretext task, as the model can easily reconstruct the signals given more visible channels, and thus its representations are not as useful. Interestingly, mask ratio = 0.9 doesn't result in a large drop in performance, unlike [12]. This suggests that higher masking ratios may be used for EEG data, as it results in fewer tokens during the encoding state, which could absorb data within more configurations for pre-training.

## C.3 Impact of Masking Strategy

Table 9: Results from variants of different masking strategy

| Random Masking | Regional Masking | Downstream Accuracy | Reconstruction loss |
|:---:|:---:|:---:|:---:|
| 0.3 | 0.7 | 94.49 | 8.26 |
| 0.5 | 0.5 | **94.61** | **8.07** |
| 0.7 | 0.3 | 93.67 | 8.12 |

In this section, we investigate the impact of the masking strategy for **MMM** on the SEED dataset. Results are shown in Table 9. The appropriate mix rate of mask strategies can bring better representations.

## C.4 Impact of depth of encoders and decoders

Table 10: Results from variants of different transformer layers

|  |  | Decoder | | |
|:---:|:---:|:---:|:---:|:---:|
|  |  | 4 | 6 | 8 |
| **Encoder** | 4 | 93.86 | 93.99 | 94.55 |
|  | 6 | **94.92** | 94.61 | 94.33 |
|  | 8 | 94.17 | 94.52 | 93.95 |

In this section, we investigate the impact of the depth of encoder and decoder in **MMM** on the SEED dataset. We use encoders and decoders with different numbers of transformer layers and report the results in Table 10

To be noticed our setting in main experiment should follow the [4] for a fair comparison, which has 6 depth for encoders and 6 depth for decoders. From a horizontal perspective, the deeper decoder can help the overall auto-encoder well model the EEG data, but too many layers would take over the difficulty and understanding of the encoders. From a vertical view, the deeper encoders can bring more understanding from the pre-training data, but too many layers would bring the overfitting problem.

## C.5 Impact of different components

To further scrutinize the design of different components, we conduct a simple but effective ablation study as shown in Table 11. To be noticed the decrement brought by Multi-stage pre-training has been explained in Section 4.7.

Table 11: Ablation study on different components of MMM on the SEED dataset. (✗) means the component is missing

| Multi-dimensional Positional Encoding | Multi-level Channel Hierarchy | Multi-stage Pre-training | SEED |
|---|---|---|---|
| ✗ | | | 95.12 |
| | ✗ | | 91.93 |
| | | ✗ | 95.28 |
| | | | 95.15 |

## D  Broader Impact

Electroencephalogram (EEG) is a physiological signal widely used in clinical practice and brain-computer interfaces. Instead of fitting models with limited data within a task, our method exploits abundant, topology-agnostic, and cross-tasks unlabelled EEG data. Our pre-trained model can draw unified and meaningful representations that are beneficial for various downstream tasks. Neurologists and engineers could make good use of such benefits in diagnosis and BCI applications.

While our work of EEG pre-training doesn't aim to handle a specific sensitive task with negative social impacts, the technique might possibly be misused in the future with privacy concerns, e.g., reading one's mind from EEG signals without acknowledgment. However, we believe that we're moving in the right direction toward the target of building an EEG foundation model to understand EEG signals better. This is essential to brain disease diagnosis like seizures, Parkinson's disease, etc., and is life-saving.

## E  Limitation

Our method is designed to provide a unified solution to different EEG configurations. While it requires a unified modeling strategy on both spatial and temporal dimensions, our focus is more on the spatial information, and we handle the temporal information with a non-parametric solution. Our framework, however, is also capable of different kinds of temporal modeling strategies, and we believe the results can be further boosted with a more advanced technique to handle temporal information better. Also, our model is currently tested only on the emotion recognition task, and we believe the framework can be easily adapted to other EEG tasks to achieve better performances with more data available for training.

