# OpenReview forum: "Learning Topology-Agnostic EEG Representations with Geometry-Aware Modeling"
_NeurIPS.cc/2023/Conference — NeurIPS 2023 poster_

### Official Review · Reviewer_v1Fw · 2023-06-23

**Soundness:** 3 good
**Presentation:** 3 good
**Contribution:** 3 good
**Rating:** 8
**Confidence:** 4

**Summary:**

The work proposes a cross-dataset cross-electrode montage deep-learning pretraining method. As a model, they use a transformer that uses the electrode coordinates as positional encodings. The transformer model then processes differential entropy features per electrode as tokens with attention across all electrodes. Additionally, it has tokens for 17 predefined brain regions with attention restricted to attention between a region and electrodes within that region as well as between regions. The transformer is pretrained using a masked autoencoder framework, where the transformer predicts differential-entropy values of randomly masked-out electrodes. Masked-out electrodes are either chosen completely randomly or all electrodes of randomly chosen brain regions are masked out. The work evaluates both cross-dataset and cross-montage experiments, showing improvements from their chosen pretraining scheme.


**Strengths:**

The approach has a clear motivation to do cross-montage pretraining, which is a potentially very useful capability. The method is straightforward and mostly easy to understand. Evaluation setup is interesting. Ablation studies are also useful.

**Weaknesses:**

I did not see how many random seeds are used to obtain the results. A lot of times accuracy differences are relatively small and may be impacted by random noise from the training process. Ideally, results should be obtained by averaging results from multiple seeds and this should be reported in the paper.

Another ablation where you add 17 artificial tokens, but they are not region-specific, so they have full attention to all electrodes and all other artificial tokens would be necessary to distinguish improvements from model capacity and region-specificness.

Table 2 Pre-trained dataset should be pre-training dataset

A lot of the fonts/text in figures is hard to read, especially Figure 1 d), but also others.

“subjects(8 f” -> missing space on p.6 l 215


**Questions:**

For attention, this was a bit confusing to me:
““As a new hierarchy is introduced and only the information from region-wise nodes is used to reconstruct the origin nodes, the unified representation must be capable of both intra-region and inter-region reconstruction. “”
As far as I understand Fig 1.d) there is attention from all electrodes to all electrodes, from all regions to all regions, and from any region to all electrodes of that region? Sentence above made it sound like maybe first part (all electrodes to all electrodes) doesn’t exist? Also please write this explicitly in the text in 3.3.2, so it is more clear.

Regarding the positional encoding, how does it relate to the positional encoding from the VIT Vision transformer? Please do mention also in the text where it is different or similar.

In Figure 3, all scalp plots of the original column look very similar to me. Could one find a more diverse set of examples?

**Limitations:**

Ablation study as written above might be more helpful to understand contribution of different parts.

---

> ### Author Rebuttal · Authors · 2023-08-10
>
> Thanks for your comments and suggestions, which are valuable for enhancing our paper. The following are responses to individual concerns:
> 1. **Random seeds (W1)**: We follow the previous work [1,2] to perform subject-dependent classification with five random seeds to reduce the effects of random noise. We will clarify this in the revised version.
>
> 2. **Comprehensive ablation study on region-specificness (W2)**: Thanks for your suggestion! Although we have investigated the region-specificness in spatial modeling by shuffling the attention between region-wise tokens and all electrodes (Table 1 in Supplementary file, 17 Chaos and 17 default), it is inspired that we can add 17 artificial tokens without any region specification to get another ablation. The result is shown in the following table (full attention). Our design still outperforms the one with full attention, further validating the effectiveness of region-specificness.
>     | Attention setting| Fine-tune Accuracy on SEED(%) |
>     |----------|----------|
>     | ours | 94.61 |
>     | chaos | 93.96 |
>     | full attention | 94.08 |
>
> 3. **Text and figure suggestions (W3, W4, W5)**: Thanks for the suggestions about the text and figures. We went through our manuscript and revised all the typos we could find accordingly. Besides, we polished our figures for better readability, and you can find them in the attached PDF file.
>
> 4. **Information Flow and Attention configuration (Q1)**: Our framework generally follows the encoder-decoder / encoder-classifier schema, where the multi-channel EEG data are first encoded by the encoder, then *only the region-wise nodes* are used for reconstruction / downstream task with the decoder and the classifier. The quoted sentence *"As a new hierarchy is introduced, and only the information from region-wise nodes is used to reconstruct the origin nodes, the unified representation must be capable of both intra-region and inter-region reconstruction."* is used to explain how we design such a bottleneck, where we only use the unified representation from encoders to reconstruct EEG channels in decoders. As for the attention configuration, the first part (all electrodes to all electrodes) exists in the forward process of both encoder and decoder layers, though the origin nodes have been re-masked to zeros before feeding to the decoder. We will make these more explicit in the revised version.
>
> 5. **Difference between positional encoding in ViT and ours (Q2)**: ViT incorporates three kinds of positional encoding: 1-D, 2-D, and relative. Multi-dimensional Position encoding is similar to 2-D position encoding, while the (x, y) denotes different things in these two concepts. The former use (x, y) to denote the physical location of EEG electrodes on the scalp, while the latter use (x, y) to denote a location in grids. We will clarify it in the revised version.
>
> 6. **More diverse examples (Q3)**: We will provide a diverse set of examples in the revised version, and a preview can be referred to Figure 2 in the attached PDF file.
>
> [1] T. Song, W. Zheng, P. Song and Z. Cui, "EEG Emotion Recognition Using Dynamical Graph Convolutional Neural Networks," in IEEE Transactions on Affective Computing, vol. 11, no. 3, pp. 532-541, 1 July-Sept. 2020, doi: 10.1109/TAFFC.2018.2817622.
>
> [2] Rui Li, Yiting Wang, Wei-Long Zheng, and Bao-Liang Lu. 2022. A Multi-view Spectral-Spatial-Temporal Masked Autoencoder for Decoding Emotions with Self-supervised Learning. In Proceedings of the 30th ACM International Conference on Multimedia (MM '22). Association for Computing Machinery, New York, NY, USA, 6–14.

---

> > ### Comment · Reviewer_v1Fw · 2023-08-11
> >
> > Thanks for the responses and additional effort put into strengthening the contributions of the study. Due to the additional ablations and additional TUEG study, I increase my score to strong accept.

---

> > > ### Author Response · Authors · 2023-08-13
> > >
> > > we greatly appreciate the reviewer's positive response to our revision and we will polish the manuscript accordingly. Again, we sincerely thank you for raising the rating score regarding our paper.

---

### Official Review · Reviewer_JKwr · 2023-06-25

**Soundness:** 3 good
**Presentation:** 4 excellent
**Contribution:** 3 good
**Rating:** 6
**Confidence:** 4

**Summary:**

This paper aims to provide new architecture that perform pre-training for scalp EEG to utilize the large-scale unlabelled data. However, one of the challenge of performing such pre-training is the different sampling channels selection & inherent structural and spatial information across different EEG datasets. Thus, this paper aims to break such boundaries by mapping different channel selection setup into a unified topology/spatial map of EEG electrodes. Based on this inspiration, the paper proposed MMM as a pre-training strategy with multi-dimensional position encoding and multi-level channel hierarchy based on a unified topology to extract representation from EEG. The paper used many experiments on emotional detection to validate their proposed method.

**Strengths:**

1. The paper aims to address a very important problem. The difference of channels selections for scalp EEG is one critical issue/limitation for current studies.
2. The proposed method is intuitive but novel, the multi-scale channel hierarchy is carefully and nicely designed. It is a valid/meaningful contribution to the community.
3. The proposed method achieves adequate performance improvements over the benchmark.
4. The writing and organization is clear. The paper is enjoyable to read.

**Weaknesses:**

Minor concerns below:

1. Aside from "montage (the number and the places of electrodes placed on the scalp) and sample rate" as mentioned in line 24-25, another issue that prevents methods from being transferrable to another dataset for scalp EEG might be the type of electrodes. e.g. wet electrodes v.s. dry electrodes, and the corresponding domain shifts of collected signals. While this is not the major issue to be tackled in this work, it'd be great if the authors can mention/discuss this factor (and other possible factors) in either the introduction or the conclusion section.

2. The related work section is not 100% complete. Some arguments are not accurate and could use some edits.
- Aside from [6] and [7], https://arxiv.org/abs/2007.04871 [Subject-Aware Contrastive Learning for Biosignals] also explored spatial augmentations for EEG data.
- "To the best of our knowledge, there is no method trying to use cross-dataset EEG as their pre-training corpus.", in [BENDR: Using Transformers and a Contrastive Self-Supervised Learning Task to Learn From Massive Amounts of EEG Data] section 2.1.1 "This also means that these data should include multiple different recording hardware and configurations." Although they eventually used TUEG dataset as the pre-training dataset (which is defaulted to be the same configuration, but due to the TUH data collection pipeline the actual data contain many variations), so I am not sure if this should be counted as "cross-dataset pre-training", this should be discussed.
- Overall, the section "Self-supervised learning for EEG" should include more related works. The authors should spend more efforts doing the literature search on this front.

3. The experimental details could be more comprehensive, some of the details should be included within the main text.
- Is the train/validation/test split randomly created or based on different subjects? Correct me if I am wrong, but I did not find related details in both the main text and the appendix. Is there k-fold cross-validation for different subjects? SEED is a relatively small dataset, so I'd expect such a dataset manipulation would not be too computationally expensive.

4. Minor writing issues. e.g. line 98 "It's" --> "It is".

**Questions:**

The experiments focused on emotional detection as the major task. While in EEG, sleep staging datasets contain much more abundant unlabelled (or labelled, in some cases) publicly available datasets that could be used for pretraining. Have the authors considered that dataset, as the proposed method is supposed to work well w cross-dataset EEG?

**Limitations:**

The authors should discuss more limitations of the work.

---

> ### Author Rebuttal · Authors · 2023-08-10
>
> Thanks for your comments and suggestions, which are valuable for enhancing our paper. We feel excited that our paper provided a pleasant reading experience and that our contributions are well recognized.
> Responses to your concerns and questions are hereby presented:
> 1. **Discussion on more transferring issues (W1)**: Good points! It is inspired to consider the type of electrodes. Besides that, inconsistent time scope and feature pre-processing are all factors that prevent transfer between tasks. These are all the core challenges in front of the EEG pre-training area, and we will add the discussion to the limitation section in the revised version. While these challenges are beyond the scope of the paper, we hope this work can provide a powerful tool to tackle the cross-montage problem and inspire the related community.
>
> 2. **Additional related work about spatial modeling with EEG (W2.1)**: Thanks for the suggestions. [Subject-Aware Contrastive Learning for Biosignals] incorporates time, frequency, and spatial augmentations on EEG data while using them on contrastive learning. The method also works on a fixed channel set, not faced with cross-montage problems. We will cite it with a few discussions.
>
> 3. **Additional related work about self-supervised training with EEG (W2.2)**: Thanks for the suggestions. BENDER work on cross-dataset pre-training by selecting a 19-channel configuration, based on which they pad missing channels to 0 and drop surplus channels during the data preparation stage. Unlike our work, they have not yet tackled the cross-montage problem from the model's perspective.
>   When faced with 62-channel data (SEED series in our experiment), BENDER will only keep the 19 channels in the 10-20 system and ignores the rest. This will lead to huge information loss, and our method is working on utilizing full spatial information faced with different channel configurations. We will cite BENDER and add a few discussions in the revised version.
>
> 3. **Experiment settings (W3)**: We follow the experiment setting of the previous work [2, 3], where the results are from the average results for each patient. It is noticed that all patients share the same number of samples, and samples in the first nine trials are treated as training samples, while samples in the last six trials are treated as test samples. The detail of the dataset can be referred to SEED [4], and we will clarify the details of the dataset in the revised version.
>
> 4. **Text suggestions (W4)**: Good catch! We went through the whole manuscript and revised all the typos we found.
>
> 5. **Larger dataset for pretraining (Q1)**: We have handled the cross-montage problem. But it is still non-trivial to do broader cross-task transitions because of challenges mentioned in Response **Discussion on more transferring issues**, such as inconsistent time scope, task-specific feature pre-processing, etc. Some of these challenges are beyond the scope of this paper. Despite the challenges, we still manage to pre-train our model on a large-scale EEG dataset, the TUEG dataset [1]. To be noticed, we maintain a similar experiment setting for the pre-training and fine-tuning, except for pre-training with 21 channels in the 10-20 international system. The fine-tuning result on SEED is shown in the following table. Even though there are huge gaps in sampling devices, subjects' physiological status, and so on, we outperform the model pre-trained on the downstream dataset by bringing massive pre-training data.
>
>     | Pre-training dataset  (pre-training epoch)| Fine-tune Accuracy on SEED(%) |
>     |-----|-----|
>     | random initialization | 94.59 |
>     | TUEG (1 epoch) | 94.85 |
>     | TUEG (2 epoch) | 95.11 |
>     | SEED | 95.15 |
>     | TUEG (3 epoch) | 95.29 |
>
> [1] Harati A, Lopez S, Obeid I, et al. The TUH EEG CORPUS: A big data resource for automated EEG interpretation[C]//2014 IEEE signal processing in medicine and biology symposium (SPMB). IEEE, 2014: 1-5.
>
> [2] T. Song, W. Zheng, P. Song and Z. Cui, "EEG Emotion Recognition Using Dynamical Graph Convolutional Neural Networks," in IEEE Transactions on Affective Computing, vol. 11, no. 3, pp. 532-541, 1 July-Sept. 2020, doi: 10.1109/TAFFC.2018.2817622.
>
> [3] Rui Li, Yiting Wang, Wei-Long Zheng, and Bao-Liang Lu. 2022. A Multi-view Spectral-Spatial-Temporal Masked Autoencoder for Decoding Emotions with Self-supervised Learning. In Proceedings of the 30th ACM International Conference on Multimedia (MM '22). Association for Computing Machinery, New York, NY, USA, 6–14.
>
> [4] Zheng W L, Lu B L. Investigating critical frequency bands and channels for EEG-based emotion recognition with deep neural networks[J]. IEEE Transactions on autonomous mental development, 2015, 7(3): 162-175.

---

> ### Comment · Reviewer_JKwr · 2023-08-18
> **Response to rebuttal**
>
> Thank you for providing detailed response. I have read through the rebuttal and other reviewer's comment, and tend to remain my score the same. While the additional experiments improved the paper's contribution, I think it only better supports the original claim, which I am already convinced of.

---

> > ### Author Response · Authors · 2023-08-21
> >
> > We sincerely value the reviewer's acknowledgment of our contribution and will update the related work section accordingly.

---

### Official Review · Reviewer_JXz6 · 2023-06-26

**Soundness:** 3 good
**Presentation:** 4 excellent
**Contribution:** 3 good
**Rating:** 8
**Confidence:** 4

**Summary:**

The paper proposed an innovative approach to the pre-training of models for the EEG data. Large-scale pre-training which demonstrated great potential in CV and NLP requires a substantial amount of data. While EEG data is relatively easy to collect, their interpretation and labelling often require substantial expert efforts. Although the integration of various datasets can address this concern, the differing electrode configurations among these datasets can result in domain shifts and dimension mismatches.

The paper introduces a framework named MMM, which stands for Multi-dimensional position encoding, Multi-level channel hierarchy, and Multi-stage pre-training strategy. The novelty lies in the mapping of all EEG channel selections onto a unified topology, enabling the development of a pre-training framework to learn unified, geometry-aware EEG representations that can generalize across different EEG channel configurations.

In their approach, the authors encode the spatial information into the representation and develop a method that allows pre-training with an EEG corpus having various sensor configurations. This is achieved through the concept of region-wise tokens, which form a multi-level hierarchy learning from the EEG channels. These tokens eventually form a unified representation that can be applied to downstream tasks.

Additionally, the paper proposes a multi-dimensional position encoding technique to encapsulate geometric sensor information and a multi-stage mask strategy involving random and region-wise masks designed to enhance the robustness of hidden representations.

The authors validate their framework against a wide range of state-of-the-art methods on EEG emotion recognition tasks. Their experimental results indicate that the proposed method not only achieves state-of-the-art performance but also exhibits a strong ability to transfer across different datasets, even those with differing montages. The author also conducted an ablation study, underscoring the validity and robustness of each proposed component, contributing to the MMM framework's overall effectiveness.

**Strengths:**

1. This manuscript makes an innovative contribution to the field by addressing the crucial issue of sensor configuration heterogeneity, which is a significant barrier to enabling large-scale pretraining using multiple EEG datasets. The novelty of the approach opens a plethora of opportunities for future research in EEG decoding and is commendable.

2. The presentation is excellent, with cohesiveness that makes the paper feel complete and thorough. All elements, from figures to text, complement each other well, creating a pleasant reading experience. The figures are particularly well designed, effectively illustrating the complex method in an understandable way. The choice of colour scheme enhances the visual appeal and readability of the work.

3. The manuscript's strength also lies in the clear articulation of motivation and contributions in the introduction. The narrative throughout the methodology section is seamless, explaining the proposed method with great clarity and depth.

4. The rigorous experimental validation of the findings is highly appreciated. The authors included comprehensive experiments covering classification tasks with various state-of-the-art methods based on RNN, CNN, and GNN. The work's transferability across different datasets and electrode configurations is also well demonstrated. In addition, the authors performed a detailed ablation study to validate the significance of each proposed component, further strengthening the work.

5. The inclusion of qualitative results, such as the topographic plot of the reconstructed data, is valuable. The results demonstrated the superior performance of the proposed MMM method in reconstructing missing channels at high mask ratios.

6. Overall, this paper makes a significant contribution to the field and is likely to inspire future work. The quality of presentation, the novelty of the approach, and rigorous experimentation all stand out as notable strengths of the work.

**Weaknesses:**

1. While the overall design and information presentation in Figure 1 is commendable, it might be beneficial to consider further optimizing the text sizes for enhanced readability. There are noticeable empty spaces that could potentially be utilized to enlarge the text, improving the figure's overall clarity and effectiveness.

2.  The manuscript could benefit from additional clarification regarding the reconstruction experiment, particularly in terms of the masking method employed. Are global random masking or regional masking methods used, or perhaps a combination of both? Exploring potential performance differences between the two masking methods would be an insightful addition. Furthermore, it would be beneficial to provide a comparison of the Mean Squared Error (MSE) between different masking methods to better understand the relative performance of the proposed method.

3. When considering the transferability experiments across different datasets and montages, it might be prudent to include the state-of-the-art methods that were employed in the regular classification task. These methods can be trained on the common channels between different datasets, aligning with conventional practices in the literature. Additionally, incorporating established EEG transfer learning techniques as comparisons for these experiments could offer a more comprehensive view. This inclusion will make the results more comparable to existing literature and potentially broaden the generalizability of the findings.

**Questions:**

1. For the Reconstruction experiment, is the mask uses global random masking or regional masking?

2. Will the code be made available?

**Limitations:**

please refer to the weaknesses

---

> ### Author Rebuttal · Authors · 2023-08-10
>
> We appreciate your comments and suggestions that truly enhanced the quality of our paper. We are cheerful that we have provided you with a pleasant reading experience, and our contribution is well recognized.
>
> Responses to your concerns are presented as follows:
> 1. **Text size of Figure 1. (W1)**: Thanks for your suggestion, Figure 1 will be revised in the next version, and a preview is available in the attached PDF file.
> 2. **Details of multi-stage mask strategy (W2, Q1)**: The global random and regional masking methods are used alternatively during pre-training, whose details are in the supplementary (Line 20-23). We will clarify this setting in the manuscript for higher readability.
> Moreover, thanks for the suggestion on adding the ablation study that uses MSE to measure different masking strategies while we are now using downstream task performances. (Supplementary Material section 3.3). The results are shown in the following table and confirm our statement that the appropriate mix rate of mask strategies can bring better representations (Line 90 in Supplementary Material). We will add these experiments to the revised version of our supplementary file.
>     | Ratio of Random Masking |Ratio of Regional Masking| Mean Square Error($\times10^{-4}$) |
>     |----------|----------|----------|
>     |0.3|0.7|8.26|
>     |0.5|0.5|8.07|
>     |0.7|0.3|8.12|
>
> 3. **Baseline trained on the common channels (W3)**: It is inspired to train models on the common channels to avoid inconsistent channel configurations. Previous work [1] has investigated the effectiveness of focusing on common channels. We also conduct more experiments with MAE, MMM, and DGCNN, as shown in the following table. To simplify the table, the [Pre-train, Fine-tune, Test] columns denote the number of channels used in the corresponding stage. The results show that the information loss caused by focusing on the common channels can not be ignored. With more tasks, the common channel would become less, and more information loss would occur. That is one of the motivations for our method, which can handle different channel configurations.
>     | Method |Pre-train|Fine-tine/train |Test| Accuracy (%)|
>     |----------|----------|----------|----------|----------|
>     |MMM|/|62|62|93.76|
>     |MMM|62|62|62|94.61|
>     |MMM|32|62|62|93.97|
>     |MAE|/|62|62|87.89|
>     |MAE|/|32|32|83.25|
>     |DGCNN|/|62|62|90.04|
>     |DGCNN|/|32|32|83.25|
>
> 4. **Code availability (Q2)**: We will publish both the code and the pre-trained model upon acceptance.
>
> [1] Kostas D, Aroca-Ouellette S, Rudzicz F. BENDR: using transformers and a contrastive self-supervised learning task to learn from massive amounts of EEG data[J]. Frontiers in Human Neuroscience, 2021, 15: 653659.

---

> > ### Comment · Reviewer_JXz6 · 2023-08-18
> >
> > Thanks for the clarifications and responses to my comments.

---

> > > ### Author Response · Authors · 2023-08-21
> > >
> > > We greatly appreciate the reviewer's positive feedback and recognition of our efforts.

---

### Official Review · Reviewer_tt92 · 2023-07-05

**Soundness:** 2 fair
**Presentation:** 1 poor
**Contribution:** 2 fair
**Rating:** 7
**Confidence:** 4

**Summary:**

This work seeks to improve classification tasks over EEG brain data using pre-training. There exist many EEG datasets, but not all datasets use the same montage format. To leverage all the datasets together, this work presents an approach for learning generic representations of EEG data. This approach proceeds in two stages. In the first stage, EEG data is given as input to a Transformer-based autoencoder. The bottleneck for this autoencoder is a small set of region-nodes, which are meant to represent the regions of the brain. In the second stage, the representations for these region nodes are used for fine-tuning on a classification task. The benefit of this method is that the region representation is agnostic to montage format.

**Strengths:**

- This work sets out to address an important question: how do we handle the fact that there is a lot of EEG data out there, but not all of the same montage format? Can we pre-train over all this data at once?
- The presented method is a reasonable first approach to this problem.

**Weaknesses:**

- My main concerns are clarity and the significance of the results
- The aim of this work is to show the ability to transfer between different datasets, but as described in 4.1, all the datasets considered seem to have a very similar source. The most different datasets, the Lite datasets, are created by extracting data from the other datasets being considered. It would strengthen this work if a wider variety of sources was considered. Or perhaps it could be better explained why the present datasets are more different than they might appear.
- The results presented in 4.5 show inconsistent and small improvement when transferring between montages.
- I have some doubts about the underlying theory in this work. It's claimed that there is a hierarchy in the representation, because independent region nodes are given as input to the model. But there is nothing about the architecture that constrains region information to pass through the region nodes.
- There are many small typos throughout. For example, sentence 1 of the abstract should probably read: "Large-scale pre-training has shown great potential to enhance models on downstream tasks in vision and language". Sometimes these typos impede clarity. For example, line 176 refers to a "symmetry stack", and it is unclear whether this denotes a specific type of transformer, or if it just means that the decoder is symmetrical to the encoder. Lines 38 and 39 have similar typos that make reading difficult. I definitely don't mean to pick on the English writing, and overall the sentences are mostly clear, but I think the clarity would be greatly improved if it was given a close reading for typos.
- The ablation study doesn't contain an experiment where the positional encoding is ablated. It would be useful to see this, since it's a key part of the method.
- Similarly, no experiment is done where Multi-level channel hierarchy is the only component missing
- I have other concerns about clarity (see questions below)

**Questions:**

- In general, how do you think this method would perform in the case where you have two datasets, but the set of regions present in each dataset is different? For example, dataset A might contain recordings for regions 1, 2, and 3. And Region B might only contain recordings on regions 4, 5, and 6.
- Can you say briefly what the main difference between MAE and MMM is?
- On line 151, it mentions the (x,y) coordinates of the nodes. What are these coordinates relative to? Every scalp is different, so were projections made to a common scalp?
- How is the Lite series created? How are the 32 channels selected? Are they the same 32 channels for every subject?
- On line 233, why does it say that the method only uses one-tenth of the information?
- I'm not sure how to understand the experiments in section 4.2? Are these the results when training is restricted to a single subject? Which subject?
- Why are there two seemingly identical models in Table 1? I'm looking at the third-from-last and second-from-last rows?
- In table 1, what is the standard deviation taken over?
- I'm not sure how to read the text in section 4.4 and table 3. What does the "SEED (Fine-tune)" row denote? Aren't the other models also finetuned on SEED?
- On line 261 it says that "We pre-train models using the Lite series dataset", but then table 3 seems to suggest that different pre-training datasets are used. For what models was the Lite series used?
- The text in 4.4 makes comparisons to the "model trained from scratch" but all the rows in the table show deltas with respect to "SEED (Fine-tune)." I suggest showing the deltas with respect to the model trained from scratch.
- On line 265, it says that the performance drop is due to the fact that SEED-IV-Lite only has 32 channels. But isn't this the case for SEED-Union-Lite dataset, which shows an increase in performance?

**Limitations:**

Limitations and potential social impacts are not discussed. But for this work, I think this is alright.

---

> ### Author Rebuttal · Authors · 2023-08-10
>
> Thanks for your valuable suggestions, which help us enhance the clarity of this paper and make it more understandable for the wider community. We reviewed your suggestions, and our manuscript to ensure all typos, vague descriptions, and unclear settings we can find (W1, W5, Q4, Q6, Q8) are properly addressed in the revised version. Please kindly refer to the global rebuttal for more details.
>
> For individual concerns, we respond as follows:
>
> 1. **Wider variety of sources (W2)**: SEED and SEED-IV contain different subjects and stimulating materials, which are non-trivial within the EEG area. SEED and SEED-Lite have different montages. Although they're designed for the same task, as we know, there is no relevant work on these two kinds of transition.
> Despite challenges, we pre-train our model on a large-scale EEG dataset, the TUH EEG Corpus [2]. We maintain a similar experiment setting for the pre-training and fine-tuning, except for pre-training with 21 channels. Please kindly refer to the global response for details and results.
>
> 2. **Inconsistency and small improvements in section 4.5 (W3)**: The ablation study is based on the setting, pre-training on SEED / SEED-Lite, and fine-tuning on SEED. For inconsistency, it suggests that the multi-stage mask strategy aids in enhancing the model's generalizability across various datasets (Lines 282-285). As for the number, a 3.88% downstream accuracy increment is shown for MMM pre-trained on SEED-lite with full components to the one with no component. We beg to differ in that it is not a small improvement.
>
> 3. **Hierarchy and constraints (W4): Hierarchy consists of region-wise nodes**, EEG channels, and their communications. We have constrained the region-wise node not to interact with EEG channels out of its region. Compared with constraints on cross-region channel interaction, the scheme in the paper can model the EEG signal more efficiently without smashing the hierarchical structure.
>
> 4. **Typos (W5)**: Thanks again for pointing out the typos. The phrase "symmetry stack" means "symmetrical to the encoder".
>
> 5. **Comprehensive Ablations (W6, W7)**: We validate the effectiveness of positional encoding by comparing the second row in Table 4. A similar comparison can be made between the second and third rows for multi-level channel hierarchy.
> Adding multi-stage pre-training without a multi-level channel hierarchy is not very sensible. However, we conduct additional experiments for rigor, as shown in the table.
>
>   |Multi-dimensional PE | Multi-level Channel Hierarchy| Multi-stage Pre-training| Accuracy on SEED (%) |
>   |-|-|-|-|
>   |1|0|1|91.93|
>   |0|1|1|95.12|
>   |1|1|1|95.15|
>
> 6. **Non-overlapping case (Q1)**: It is right to worry that our method cannot handle this extreme case without overlapped channels. But most datasets have channels uniformly distributed on the scalp, such as TUEG [2], Sleep-edf [3], etc. And the majority use subsets of the 10-20/10-10 international system [1]. Even though the extreme case happens, we could practically introduce the C dataset, which overlaps A and B.
>
> 7. **Main difference between MAE and MMM (Q2)**: Table 4 has shown that the main difference between MAE and MMM comes from a multi-level channel hierarchy, i.e., 1) replacing the class token with region-wise tokens for a spatial-aware representation (Line 168-169). 2) reconstruction with region-wise tokens instead of the visible channels to enforce the capability of the representation (Line 176-178).
>
> 8. **Coordinates for position encoding (Q3)**: (x,y) coordinates are relative to the channels' position in the 10-10 international system [1] (Line 152). Thanks to the unified system, most EEG-related research does not worry about different scalps.
>
> 9. **Creating Lite series (Q4)**: Lite series is created by evenly selecting 32 channels from 62 channels. The selected channels remain the same across subjects.
>
> 10. **One-tenth information (Q5)**: MV-SSTMA uses ten-timestep information, while we only use one timestep (shown in Table 1 "Window Size").
>
> 11. **Subject-dependent Classification (Q6, Q8)**: We follow the previous work [4, 5] to perform subject-dependent classification. Each subject has 15 recorded trials with emotion labels. The first 9 are for training, and the rest are for testing. Each subject has its own training and test sets, and the reported results are the average for each subject. The standard deviation is taken over the result from each subject.
>
> 12. **Seemingly identical models (Q7)**: The difference is the dagger symbol (†), representing excluding the worst-performing session out of three for each subject when calculating the accuracy (refer to the caption of Table 1). This brings fair comparisons following different previous settings, i.e., results with/without the dagger symbols are directly comparable.
>
> 13. **Explanation on Section 4.4 (Q9~11)**: Section 4.4 compares models with different aspects: 1) method (MMM or MAE). 2) fine-tuning setting (fully or partially fine-tuning). 3) pre-training dataset (none, SEED, Lite series). The "SEED (Fine-tune)" row denotes the model pre-trained and fine-tuned on SEED. The rest denotes the model partially fine-tuned. i.e., the parameters of the encoder are frozen, and only the last MLP is tuned (Line 262).
> We will take your advice to show the deltas w.r.t model trained from scratch for better presentation.
>
> 14. **Line 265 (Q12)**: SEED-Union-Lite combines SEED-Lite and SEED-IV-Lite (Line 223-224) and more data bring better performance.
>
> [1] Homan R W, 1988. The 10-20 electrode system and cerebral location
>
> [2] Harati A et al., 2014. The TUH EEG CORPUS: A big data resource for automated EEG interpretation
>
> [3] Kemp B et al., 2018. Sleep-edf database expanded
>
> [4] Song T et al., 2020. EEG Emotion Recognition Using Dynamical Graph Convolutional Neural Networks
>
> [5] Li R et al., 2022. A Multi-view Spectral-Spatial-Temporal Masked Autoencoder for Decoding Emotions with Self-supervised Learning.

---

> > ### Comment · Reviewer_tt92 · 2023-08-18
> > **Response to Authors; increased score**
> >
> > I thank the authors for their thorough response. I will increase my score to a 7 for the following reasons:
> > - (W1) I had misunderstood the contents of SEED and SEED-IV. I now understand that they contain different subjects.
> > - (W2) Fair enough. It is a modest but significant improvement.
> > - (Q1, Q3) Given these facts that commonly hold for EEG data, I do not think sparsity or coordinate consistency pose a serious weakness.
> > - (W4) I had misunderstood how nodes interact with each other. This seems like a reasonable scheme.
> >
> > I still recommend that the paper be given a close reading for typos.

---

> > > ### Author Response · Authors · 2023-08-21
> > >
> > > We are pleased with the positive feedback and will address the typos in our revised version. Thank you for raising the rating score.

---

### Official Review · Reviewer_rN6e · 2023-07-06

**Soundness:** 2 fair
**Presentation:** 2 fair
**Contribution:** 3 good
**Rating:** 6
**Confidence:** 4

**Summary:**

This paper explores the application of large-scale pre-training techniques to scalp electroencephalogram (EEG) data. They leverage the abundance of unlabeled EEG data and address challenges related to sampling channel selection, structural information, and spatial information. To enable cross-dataset EEG pre-training, they propose a unified topology that maps different channel selections. They introduce MMM, a pre-training framework with multi-dimensional position encoding, multi-level channel hierarchy, and a multi-stage pre-training strategy based on the unified topology. Experimental results demonstrate significant improvements over previous state-of-the-art methods on benchmark datasets for emotional recognition.

**Strengths:**

- The authors present a novel pretraining method for cross-dataset pretraining on EEG signals, which proves to be an effective method for improvements for emotion recognition. This framework seems to be promising in providing researchers in the EEG space with larger datasets for pretraining.

**Weaknesses:**

- Although the authors present a pretraining method for cross-dataset EEG signals, they only combine the SEED and SEED-IV datasets. It would be extremely enlightening if the authors combined N > 2 datasets to see if their pretraining framework generalizes across them.
- I am a bit confused on how the emotion recognition accuracy can be so high while the loss seems to not converge well and topological maps (Figure 3) seem to not be reconstructed well. Would the authors be able to clarify this phenomenon?

**Questions:**

- I am curious on whether this pretraining method can work on different downstream tasks, such as EEG to text translation. Have the authors considered this?
- The improvements/decrease in performance of adding Multi-stage Pre-training is seen for SEED and SEED-Lite datasets. How computationally expensive is the multi-stage pre-training? I am trying to see if this tradeoff is worth the slight, possible improvement it may give.
- For Figure 3, the topological maps created by MAE seems to be quite different from the ground truth. Additionally, the topological maps created by MMM seems to be slightly better than MAE but still pretty different from the ground truth. Figure 4 also seems to present MMM having a better loss reconstruction than MAE, however, both seem to converge at a mask ratio of 0.5. I also want to note that Table 2 in the supplementary material, the random masking ratios seems to not have much difference for the performance. Have the authors considered trying different loss functions to observe distinct behaviors in loss?
- Please correct me if I am wrong, but it seems like the paper does not clarify whether the results in the paper are from the averaged EEG signals across all patients or an average of the results for each individual patient. Would the authors be able to kindly clarify a bit more?

**Limitations:**

The authors adequately addressed the limitations of the paper.

---

> ### Author Rebuttal · Authors · 2023-08-10
>
> We appreciate your comments and suggestions that truly enhanced the quality of our paper.
> 1. **Cross-task challenge (W1, Q1)**: We have handled the cross-montage problem. But it is still non-trivial to do broader cross-task transitions because of challenges such as inconsistent time scope, task-specific feature pre-processing, etc. Some of these challenges are beyond the scope of this paper.
>   Despite this, we still managed to pre-train our model on a large-scale EEG Corpus, the TUH EEG Corpus (TUEG) [1], designed for several medical tasks, including seizure detection, slow wave detection, etc., and fine-tune it on the SEED dataset. To be noticed, we maintain a similar experiment setting for the pre-training and fine-tuning, except for pre-training with 21 channels in a 10-20 system. The fine-tuning result on SEED (62 channels) is shown in the following table. Even though there are huge gaps in sampling devices, subjects' physiological status, and so on, we outperform the model pre-trained on the downstream dataset by bringing massive pre-training data (~1.7T).
>   This not only suggests that our pre-training framework can generalize among >2 datasets but also show its effectiveness in leveraging large-scale resources with different channel configurations to boost the performance further.
>
>   | Pre-training dataset (pre-training epoch) | Fine-tuning Accuracy on SEED (%) |
>   |-----|-----|
>   | random initialization | 94.59 |
>   | TUEG (1 epoch) | 94.85 |
>   | TUEG (2 epoch) | 95.11 |
>   | SEED | 95.15 |
>   | TUEG (3 epoch) | 95.29|
>
>
> 2. **Regarding Figure 3 and Figure 4, Reconstruction convergence (W2, Q3)**: To clarify, reconstruction error (Figure 3, 4) is used to measure the reconstruction quality during the *pre-training* stage, and the emotional recognition accuracy is used for downstream task performance after *fine-tuning*. Despite expected correlations between them, the pre-training score does not directly account for the downstream task performance.
>   As for the convergence problem, Figure 4 demonstrates the reconstruction loss at different experiment settings during the pre-training stage. Combined with Figure 3, we are showing MMM has a more robust EEG understanding ability than MAE by verifying their abilities to reconstruct corrupted EEG segments of different mask ratios, and all the reported results are from converged checkpoints. The reviewer might use 'not converge well' to describe the high loss. We further use the R2 score to demonstrate the reconstruction quality in Figure 3 of the attached PDF file in the global response. Moreover, because of the possible gap between pre-training and downstream tasks, we assume that over-chasing low reconstruction loss would not bring better downstream performance.
>
> 3. **More downstream tasks (Q1)**: Good point! Our method should work on different downstream tasks with different channel configurations. However, it is not yet trivial to do broader cross-task transitions because of the challenges such as inconsistent time scope, feature pre-processing, etc. Take the EEG-to-text translation task as an example. Previous work [4] uses eight frequency bands to extract EEG features on ZuCo [5, 6] dataset with inconsistent lengths between EEG segments and textual sentences, while our experiments on SEED use five frequency bands with fixed lengths of time per sample.
>   Nevertheless, such EEG tasks are still in the scope of EEG understanding, which will be a good future work with the help of our unified hierarchical representation.
>
> 4. **Details about Multi-stage pre-training (Q2)**: Multi-stage pre-training is a strategy that alternatively uses different masks in different iterations, and the global random mask and region-wise mask have the same computation cost. Since the total number of training iterations is kept the same during pre-training, there is no trade-off for additional computations.
>
> 5. **Regarding Table 2 in the supplementary material (Q3)**: The result of Table 2 in the supplementary material helps confirm our assumption that the different pre-training setting (mask ratio) does not bring too much difference in the downstream task (See Response 2: Reconstruction convergence), i.e., with the help of region-wise tokens, our framework is more robust to high mask ratios.
> And we will consider your suggestion of bringing more loss functions in future work.
>
> 6. **Way to get results (Q4)**: We follow the experiment setting of the previous work [2, 3], where the results are from the average results of all patients. We will clarify the details of the dataset in the revised version.
>
> [1] Harati A, Lopez S, Obeid I, et al. The TUH EEG CORPUS: A big data resource for automated EEG interpretation[C]//2014 IEEE signal processing in medicine and biology symposium (SPMB). IEEE, 2014: 1-5.
>
> [2] T. Song, W. Zheng, P. Song and Z. Cui, "EEG Emotion Recognition Using Dynamical Graph Convolutional Neural Networks," in IEEE Transactions on Affective Computing, vol. 11, no. 3, pp. 532-541, 1 July-Sept. 2020, doi: 10.1109/TAFFC.2018.2817622.
>
> [3] Rui Li, Yiting Wang, Wei-Long Zheng, and Bao-Liang Lu. 2022. A Multi-view Spectral-Spatial-Temporal Masked Autoencoder for Decoding Emotions with Self-supervised Learning. In Proceedings of the 30th ACM International Conference on Multimedia (MM '22). Association for Computing Machinery, New York, NY, USA, 6–14.
>
> [4] Wang Z, Ji H. Open vocabulary electroencephalography-to-text decoding and zero-shot sentiment classification[C]//Proceedings of the AAAI Conference on Artificial Intelligence. 2022, 36(5): 5350-5358.
>
> [5] Hollenstein N, Rotsztejn J, Troendle M, et al. ZuCo, a simultaneous EEG and eye-tracking resource for natural sentence reading[J]. Scientific data, 2018, 5(1): 1-13.
>
> [6] Hollenstein N, Troendle M, Zhang C, et al. ZuCo 2.0: A Dataset of Physiological Recordings During Natural Reading and Annotation[C]//Proceedings of the 12th Language Resources and Evaluation Conference. 2020: 138-146.

---

> > ### Comment · Reviewer_rN6e · 2023-08-20
> >
> > Thank you for attending to my questions. I have thoroughly read through the rebuttal and decide to raise my score to a 6. The authors have clarified my questions in an elegant manner. Thanks for the great work!

---

> > > ### Author Response · Authors · 2023-08-21
> > >
> > > We sincerely appreciate the reviewer's constructive feedback and will refine the text as suggested. Thank you sincerely for the improved rating.

---

### Author Rebuttal · Authors · 2023-08-10

Global response:
We're cheerful that the reviewers found our method novel (Reviewer rN6e, JXz6, JKwr), well motivated and reasonable (Reviewer tt92, JXz6, JKwr, v1Fw), and important (all reviewers). We're also delighted that reviewers feel reading our manuscript is pleasant (JXz6), joyful (JKwr), and easy-to-understand (v1Fw).

A major concern most reviewers share is the absence of a wider variety of sources. Even though we have handled the cross-montage problem, it is still non-trivial to do broader cross-task transitions because of some challenges, such as the type of electrodes(Reviewer JKwr), inconsistent time scope, task-specific feature pre-processing, etc. Some of these challenges are beyond the scope of this paper. Nevertheless, we still manage to pre-train our model on a large-scale EEG Corpus, the TUH EEG Corpus (TUEG) [1], and maintain the same pre-training/fine-tuning setting as in the main paper, except for pre-training models on 21 channels of EEG data. The results are shown in the following table, which further validate the effectiveness of our method on cross-montage problems.
  | Pre-training dataset (pre-training epoch) | Fine-tuning Accuracy on SEED (%) |
  |-----|-----|
  | random initialization | 94.59 |
  | TUEG (1 epoch) | 94.85 |
  | TUEG (2 epoch) | 95.11 |
  | SEED | 95.15 |
  | TUEG (3 epoch) | 95.29|

Another concern is the clarification of the paper. We thank for reviewers for their feedback. We are trying to address the concerns and polish our paper in the revised version to make our work better understandable to the wider community. Specifically:
1. We fix all the typos we found throughout the manuscript, including space before brackets, expansion of abbreviations, misspellings, etc.
2. Figure 1 now has a better text font size. Refer to Figure 1 in the attached PDF file.
3. Figure 3 now has more diverse examples. Refer to Figure 2 in the attached PDF file.
4. We add more explicit details of our experiment setting.
5. We add discussions on the remaining challenges for cross-task EEG training to the limitation section.

Finally, we would like to express our great appreciation and excitement that several reviewers recognize our promising potential to inspire the EEG community for further exploration. We want to emphasize that our work contributes to the community not only by proposing a particular model working well with the EEG emotion recognition task but also by providing a paradigm of self-supervised learning which can tackle cross-montage problems as the first step of large-scale EEG pre-training. We believe that it is worth publishing to stimulate further discussion.

[1] Harati A, Lopez S, Obeid I, et al. The TUH EEG CORPUS: A big data resource for automated EEG interpretation[C]//2014 IEEE signal processing in medicine and biology symposium (SPMB). IEEE, 2014: 1-5.

---

### Decision · Program_Chairs · 2023-09-21

**Decision:**

Accept (poster)

**Comment:**

Large-scale pretraining for EEG could significantly improve the accuracy and reliability of EEG. From the ML point of view, what it takes to perform well in this structured and somewhat idiosyncratic domain could lead to other advances. To that end, reviewers asked for various ablations to demonstrate which aspects of this work matter and for additional cross-dataset experiments. Authors provided such in the rebuttal satisfying the reviewers.

Reviewers were unanimous in their recommendation that this work should be accepted. Multiple reviewers would have liked to see the method applied to other decoding tasks. But all were satisfied with the current experiments. Cross-dataset demonstrations of the method were particularly convincing as were extensive comparisons against current methods.

EEG is a popular method to investigate the brain and additional tools to enhance it will find a wide audience. What is presented here is not the BERT-equivalent for EEG which the community can widely adopt, but it is a step in that direction.